# Guillotine Regularization: Why removing layers is needed to improve generalization in Self-Supervised Learning

**Florian Bordes**                                                    *florian.bordes@umontreal.ca*
*Meta AI Research*
*Mila, Université de Montréal*

**Randall Balestriero**
*Meta AI Research*

**Quentin Garrido**
*Meta AI Research*
*Université Gustave Eiffel,*
*CNRS, LIGM*

**Adrien Bardes**
*Meta AI Research*
*Inria*

**Pascal Vincent**
*Meta AI Research*
*Mila, Université de Montréal*
*CIFAR*

**Reviewed on OpenReview:** *https://openreview.net/forum?id=ZgXfXSz51n*

## Abstract

One unexpected technique that emerged in recent years consists in training a Deep Network (DN) with a Self-Supervised Learning (SSL) method, and using this network on downstream tasks but *with its last few projector layers entirely removed.* This *trick of throwing away the projector* is actually critical for SSL methods to display competitive performances on ImageNet for which more than 30 percentage points can be gained that way. This is a little vexing, as one would hope that the network layer at which invariance is explicitly enforced by the SSL criterion during training (the last projector layer) should be the one to use for best generalization performance downstream. But it seems not to be, and this study sheds some light on why. This trick, which we name Guillotine Regularization (GR), is in fact a generically applicable method that has been used to improve generalization performance in transfer learning scenarios. In this work, we identify the underlying reasons behind its success and show that the optimal layer to use might change significantly depending on the training setup, the data or the downstream task. Lastly, we give some insights on how to reduce the need for a projector in SSL by aligning the pretext SSL task and the downstream task.

## 1 Introduction

Many recent self-supervised learning (SSL) methods consist in learning invariances to specific chosen relations between samples – implemented through data-augmentations – while using a regularization strategy to avoid collapse of the representations (Chen et al., 2020a;c; Grill et al., 2020; Lee et al., 2021; Caron et al., 2020; Zbontar et al., 2021; Bardes et al., 2022; Tomasev et al., 2022; Caron et al., 2021; Chen et al., 2021; Li et al., 2022; Zhou et al., 2022a;b). Incidentally SSL learning frameworks also heavily rely on a simple trick

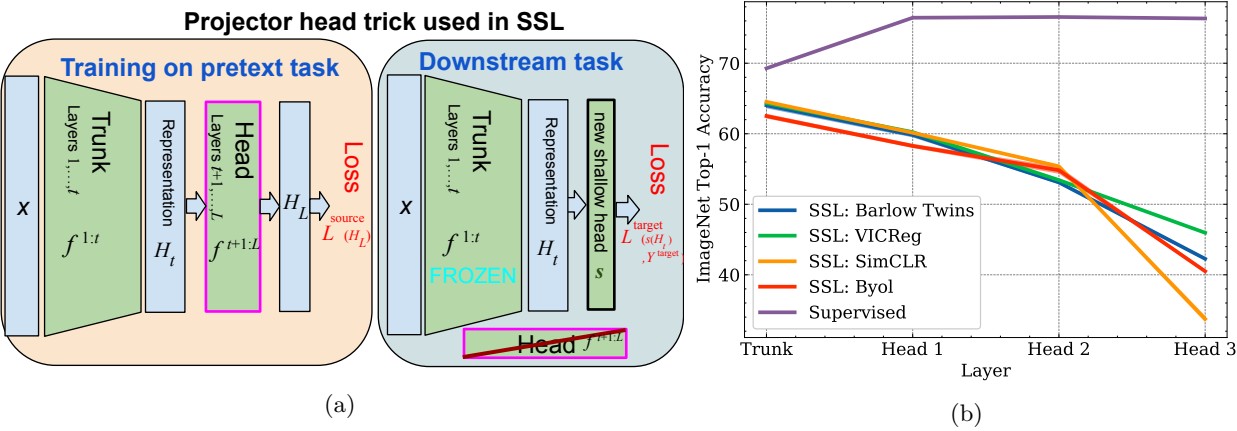

(a)                                    (b)

Figure 1: a) An illustration of projector head trick used in SSL. During training, a small neural network named the *Head* (also coined as *projector* in the SSL literature (Chen et al., 2020a)) is added on top of another deep network refereed as the *Trunk*. This *Head* can be viewed as a buffer between the training loss and the Trunk that can absorb any bias related to a ill optimisation. When using such network on downstream tasks, we throw away the Head. b) We measure with linear probes the accuracy at different layers on a Resnet50 (as Trunk) (see Figure 10 for vision transformers) on which we added a small 3-layer MLP (as Head) for various supervised and self-supervised methods. For each method, we show the mean and standard deviation across 3 runs (The std between different runs is low). With traditional supervised learning, there is a significant drop in performances when using the trunk layer instead of the last projector layer. However, when looking at self-supervised methods, the gap in performances between the linear probe trained at the trunk and projector can be as high as 30%.

to improve downstream task performances: *removing the last few layers of the trained deep network* depicted in Figure 1a. From a practical viewpoint, this technique emerged naturally (Chen et al., 2020a) through the search of ever increasing SSL performances. In fact, on ImageNet (Deng et al., 2009), such technique can improve classification performances by around 30 points of percentage (Figure 1b).

Although it improves performances in practice, not using the layer on which the SSL training was applied is unfortunate. It means throwing away the representation that was explicitly trained to be invariant to the chosen set of data augmentations, thus breaking the implied promise of using a more structured, controlled, invariant representation. By picking instead a representation that was produced an arbitrary number of layers above, SSL practitioners end up relying on a representation that likely contains much more information about the input (Bordes et al., 2021) than should be necessary to robustly solve downstream tasks.

Although the use of this technique emerged independently in SSL, using intermediate layers of a neural network–instead of the deepest layer where the initial training criterion was applied– has long been known to be useful in transfer learning scenarios (Yosinski et al., 2014). Features in upstream layers often appear more general and transferable to various downstream tasks than the ones at the deepest layers which are too specialized towards the initial training objective. This strongly suggests a related explanation for its success in SSL: does removing the last layers of a trained SSL model improve performances *because of a misalignment between the SSL training task (source domain) and downstream task (target domain)?*

In this paper, we examine that question thoroughly. We first place the SSL *trick of removing the projector post-training* under the umbrella of a generically applicable method that we call **Guillotine Regularization**. We argue that it is important to distinguish the action of removing layers during evaluation from architecture modifications because the optimal layer to use for a given downstream task is not always the backbone and could be intermediate projector's layers. Then, we explore how changes in the training optimization, training data and downstream task impact the optimal layer in both supervised and self-supervised setting. Lastly, we demonstrate that increasing the *alignment* between the pretext and downstream task in SSL decreases the need to use a projector in SSL.

To summarize, this paper's main contributions are the following:

- Since the optimal layer to use in Self-Supervised-Learning might not always be the backbone, we suggest coining the action of removing layer as a general method called Guillotine Regularization to distinguish it from the architectural modification which is the addition of a projector.

- To show through experiments that the optimal layer to cut heavily depend on the training optimization, training data and downstream task for both supervised and self-supervised models. We hope that this result will encourage the research community to run more systematic evaluations through different layers.

- The need to use Guillotine Regularization in SSL depends heavily on how the positives views are defined. When these views are aligned with the downstream-task, the optimal layer to use become closer to the last layer.

## 2 Related work

**Self-supervised learning**   Many recent works on self-supervised learning (Chen et al., 2020a;c; Grill et al., 2020; Lee et al., 2021; Caron et al., 2020; Zbontar et al., 2021; Bardes et al., 2022; Tomasev et al., 2022; Caron et al., 2021; Chen et al., 2021; Li et al., 2022; Zhou et al., 2022a;b) rely on the addition of few non linear layers (MLP) – termed *projection head* – on top of a well established neural network – termed *backbone* – during training. This addition is done regardless of the neural network used as backbone, it could be a ResNet50 (He et al., 2016) or a Vision Transformer (Dosovitskiy et al., 2021). After training, the projector is usually threw away to evaluate the model using the backbone representation. Even if Chen et al. (2020b) demonstrated that the optimal layer to use might not always be the backbone when using few labelled data, most recent works introducing new SSL methods have continued to use only the backbone for evaluation. Some works also tried to understand why a projection head is needed for self-supervised learning. Appalaraju et al. (2020) argue that the nonlinear projection head acts as filter that can separate the information used for the downstream task from the information useful for the contrastive loss. In order to support this claim, they used deep image prior (Ulyanov et al., 2018) to perform features inversion to visualize the features at the backbone level and also at the projector level. They observe that features at the backbone level seem more suitable visually for a downstream classification task than the ones at the projector level. Another related work (Bordes et al., 2021) similarly tries to map back the representations to the input space, this time by using a conditional diffusion generative model. The authors present visual evidence confirming that much of the information about a given input is lost at the projector level while most of it is still present at the backbone level. Another line of work tries to train self-supervised models without the use of a projector. Jing et al. (2022) shows that by removing the projector and cutting the representation vector in two parts, such that a SSL criteria is applied on the first part of the vector while no criterion is applied on the second part, improves considerably the performances compared to applying the SSL criteria directly on the entire representation vector. This however works mostly thanks to the residual connection of the resnet. In contrast with these approaches, our work focus on identifying which components of traditional SSL training pipelines can explain why the performances when using the final layers of the network are so much worse than the ones at the backbone level. This identification will be key for designing future SSL setups in which the generalisation performance doesn't drop drastically when using the embedding that the SSL criterion actually learns.

**Transfer learning**   The idea of using the intermediate layers of a neural network is very well known in the transfer learning community. Work like Deep Adaptation Network (Long et al., 2015) freeze the first layers of a neural network, fine-tune the last layers while adding a head which is specific for each target domain. The justification behind this strategy is that deep networks learn general features (Caruana, 1994; Bengio, 2012; Bengio et al., 2011), especially the ones at the first layers, that may be reused across different domain (Yosinski et al., 2014). Oquab et al. (2014) demonstrate that when limited amount of training data are available for the target tasks, using the frozen features extracted from the intermediate layers of a deep network trained on classification can help solve object and action classification tasks on other datasets.

Another line of work on training with random or noisy labels also studied how the use of intermediate layers improves significantly downstream performances (Maennel et al., 2020) while Baldock et al. (2021) introduced a measure of example difficulty that leverages the number of intermediate layers that are aligned towards a given prediction. In this paper, we show that SSL trained models fall under the realm of transfer learning, in consequence we can expect that all the observations made in the transfer learning literature about the use of intermediate layers are also valid for SSL. When viewing the projector SSL trick and cutting layer for transfer as a general machine learning trick to improve generalization, it's not surprising anymore that work as Wang et al. (2022); Sarıyıldız et al. (2023) have been able to show that adding a projector can also be highly beneficial for supervised training.

**Out of distribution (OOD) generalization**   Kirichenko et al. (2022) demonstrates that retraining only the last layer with a specific reweighting helps to "forget" the spurious correlations that were learned during the training. Such work emphasizes that most of the spurious correlation due to the training objective is contained in the last layers of the network. Thus, retraining them is essential to remove such spurious correlation and generalize better on downstream tasks. Similarly Rosenfeld et al. (2022) show that retraining only the last layers is most of the time as good as retraining the entire network over a subset of downstream tasks. Lastly, Evci et al. (2022) demonstrates the usefulness of using intermediate layers for OOD. Our study also confirms that Guillotine Regularization show important properties with respect to OOD generalization.

## 3   Guillotine Regularization: A regularization scheme to improve generalization of deep networks

In this section, we provide a definition for Guillotine Regularization. Then, through experiments, we show that the optimal layer to use changes significantly depending on different factors. Finally, we show that the performances at a given layer are not always correlated with the performances one can have at another layer.

### 3.1   (Re)Introducing Guillotine Regularization From First Principles

We distinguish between a **source** *training task* with its associated training set, and a **target** *downstream task* with its associated dataset[1]. It is the performance on the downstream task that is ultimately of interest. In the simplest of cases both tasks could be the same, with their datasets sampled i.i.d. from the same distribution. But more generally they may differ, as in SSL or transfer learning scenarios. In SSL we typically have an *unsupervised* training task, that uses a training set with no labels, while the downstream task can be a supervised classification task. Also note that while the bulk of training the model's parameters happens with the training task, transferring to a different downstream task will require some additional, typically lighter, training, at least of a final layer specific for that task. In our study we will focus on the use of a representation computed by the network trained on the training task and then frozen, which gets fed to a simple linear layer that will be tuned for the downstream task. This "linear evaluation" procedure is typical in SSL and aims to evaluate the quality/usefulness of an unsupervised-trained *representation*. Our focus is to ensure good generalization to the downstream task. Note that training and downstream tasks may be misaligned in several different ways.

Informally, Guillotine Regularization consists in the following: for the *downstream task*, rather than using the last layer (layer $L$) representation from the network trained on the *training task*, instead use the representation from a few layers above (layer $t$, with $t < L$). We thus *remove* a small multilayer "head" (layers $t + 1$ to $L$) of the initially trained network, hence the name of the technique. We call the remaining part (layers 1 to $t$) the *trunk*[2].

Formally, we consider a deep network that takes an input $X$ and computes a sequence of intermediate representations $H_1, \ldots, H_L$ through layer functions $f^{(1)}, \ldots f^{(L)}$ such that $H_\ell = f^{(\ell)}(H_{\ell-1})$, starting from

---

[1]Terminology pretext-training / downstream comes from SSL, while source / target is used in transfer learning
[2]head / trunk are also known as projection head / backbone in the SSL literature

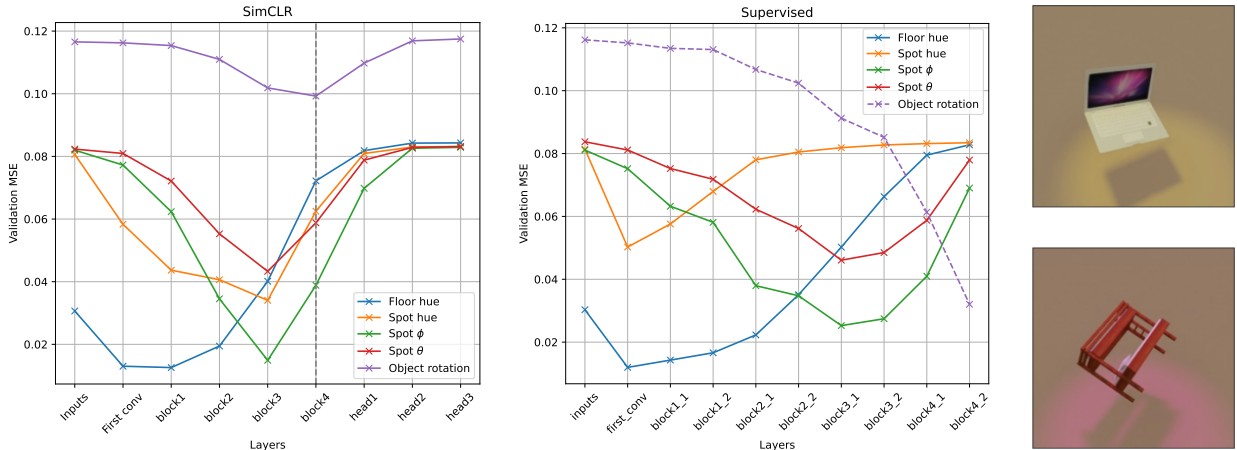

Figure 2: Training a linear regression to predict latent variables from pooled intermediate representations of a network trained with a self-supervised objective (using SimCLR) or a supervised objective (trained to predict 3D rotations of an object). The data used consists of renderings of 3d objects from 3D Warehouse (Trimble Inc) where we control the floor, lighting and object pose with latent variables, see samples on the right. The dimension of the intermediate representations increases throughout the layers and is kept constant in the head, if there is one. In the supervised setting, when looking at the Validation Mean Squared Error for object rotations prediction, the lowest error is obtained with the linear probe at the last layer of the neural networks. In contrast, the lowest error for other attributes like the Spot $\theta$ prediction are obtained with the linear probes localized 3,4 or 5 layers before the output of the networks. In the self-supervised setting, we also see that the predictor is responsible for a lot of the invariance to augmentation, and that the information is most easily retrievable before it. These results highlight the need to use Guillotine Regularization i.e removing the last layers of the neural network to generalize better on other tasks.

$H_0 = X$. The entire computation from input $X$ to last layer representation $H_L$ is thus a composition of layer functions[3]:

$$H_L = f_{\theta,\phi}(X) = (\underbrace{f^{(L)} \circ \cdots \circ f^{(t+1)}}_{\text{head } f_\phi^{t+1:L}} \circ \underbrace{f^{(t)} \circ \cdots \circ f^{(1)}}_{\text{trunk } f_\theta^{1:t}})(X)$$

The parameters $\theta$ and $\phi$ of trunk $f_\theta^{1:t}$ and head $f_\phi^{t+1:L}$ are then trained on the entire training set of examples $\mathbf{X}^{\text{source}}$ of the training task (optionally with associated targets $\mathbf{Y}^{\text{source}}$ that we may have in transfer scenarios, but will typically be absent in SSL), to minimize the training task objective $L^{\text{source}}$:

$$\hat{\theta}, \hat{\phi} = \underset{\theta,\phi}{\arg\min} \, L^{\text{source}}(f_\phi^{t+1:L}(f_\theta^{1:t}(\mathbf{X}^{\text{source}})), \mathbf{Y}^{\text{source}})$$

Then the multilayer head $f_\phi^{t+1:L}$ is discarded, we add to the trunk a (usually shallow) new head $s_w$ and we train its parameters $w$, using the training set of examples for the downstream task $(\mathbf{X}^{\text{target}}, \mathbf{Y}^{\text{target}})$, to minimize the downstream task objective $L^{\text{target}}$:

$$\hat{w} = \underset{w}{\arg\min} \, L^{\text{target}}(s_w(\underbrace{f_{\hat{\theta}}^{1:t}(\mathbf{X}^{\text{target}})}_{\text{representation } \mathbf{H}^{\text{target}}}), \mathbf{Y}^{\text{target}})$$

---

[3]Precisely, a "layer function" $f^{(\ell)}$ can correspond to a standard neural network layer (fully-connected, convolutional) with no residual or shortcut connections between them, or to entire blocks (as in densenet, or transformers) which may have internal shortcut connections, but none between them.

### 3.2 An empirical analysis of situations in which cutting layers is beneficial

There are several situations that can create a misalignment between a training and a downstream task. Here we name of few:

**Misalignment between the training (source) and downstream (target) task while using the same input data distribution.** The potential effectiveness of GR for transfer is not surprising since this technique has been used for years in the transfer learning research literature (Yosinski et al., 2014) to improve generalization across different tasks. As a simple illustration, we present Figure 2 which show how much performances on a given task can vary depending on which layer has been chosen as features extractor. In this figure, we used an artificially created object dataset in which we are able to play with different factors of variations. The dataset consists of renderings of 3D models from 3D warehouse (Trimble Inc). Each scene is built from a 3D object, a floor and a spot placed on top of the object to add lighting. This allows us to control every factor of variation and produce complex transformations in the scene. We vary the rotation of the object defined as a quaternion, the hue of the floor, and the spot hue as well as it position on a sphere using spherical coordinates. We provide more details on the dataset and rendering samples in the appendix. We observe in Figure 2 that when training a supervised model on the object rotation prediction task and evaluating the linear probe on the same task across different layers, the best results are obtained on the last layer. However, when using the same frozen neural network to predict other attributes like the Spot $\theta$, the best performances are obtained few layers before the last one. Similarly, when training with a self-supervised objective (SimCLR), we can see that the different factors of variation are most easily retrievable before the projector. This means that representations before the projector will be more versatile as they will contain information that was removed by the pretraining task. For example if our downstream task is to predict the rotation, the representation at block4 will be optimal while if the downstream task is to predict the spot hue, the representation at the block 3 will be optimal. Such results highlight the need to use Guillotine Regularization when there is a shift in the prediction task. Moreover, the optimality of a layer depends on the downstream task.

**Misalignment due to badly optimized network** It can be expected that the optimal layer to use to train a downstream task readout function might be different depending on how much the pretrained network is overfitting on the pretext task. To test this hypothesis, we train a headed supervised Resnet50 on ImageNet with two different types of optimization. The first one using only AdamW with a small learning rate of $1e-4$ without any additional regularization. The second one using SGD and the recommended hyper-parameters for a supervised training (with cycling learning rate, weight decay and momentum). In Figure 3a, we observe that the AdamW trained network that is overfitting on the classification task has readout function performances that are very close across different layers. However, when looking at the well-regularized model with SGD which does not overfit on the task, the readout performances across layers vary significantly. In a second experiment, we study more in-depth the effect of overfitting by training the Resnet50 over only a random subset of 250 classes. Then we use the remaining 750 classes as an OOD validation set that is split randomly in other subset of 250 classes. In Figure 3b, we clearly see that the training readout is overfitting on the training set while the readout performances across layers are similar on the corre-

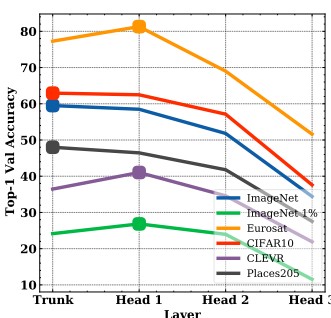

Figure 4: SimCLR: Linear probe accuracy on several downstream tasks. **The optimal layer to cut is not the same for different downstream tasks.**

.

sponding in distribution validation set (which is similar to the previous experiment over the full ImageNet). Then, we train linear probes over the OOD splits and observe that the performances are radically different from the in distribution validation set. In fact, in this instance the best layer to use for every of these split is the backbone layer whereas the best layer to use for the in-distribution split is the projector layer. **This result highlights that the optimal layer to discard can vary depending on the optimization techniques and downstream data distribution, even when the same training objective is used.**

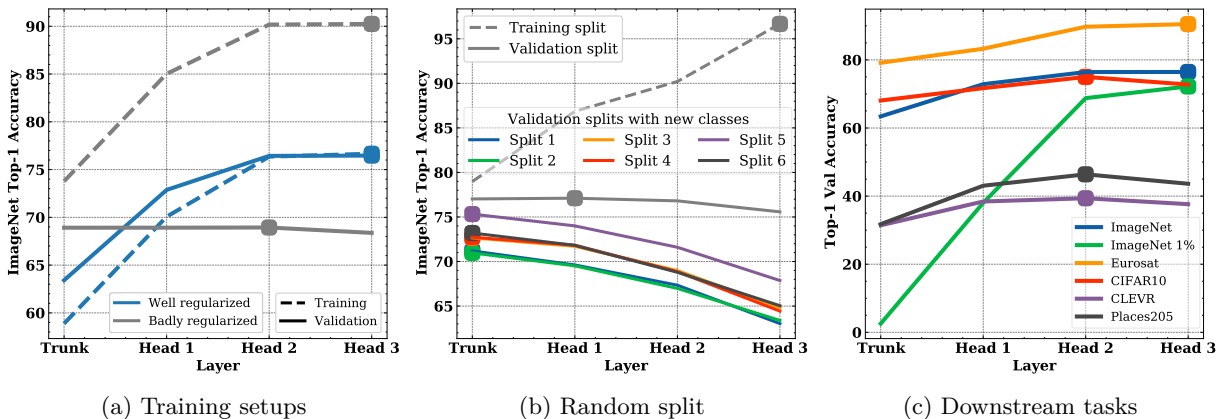

(a) Training setups      (b) Random split      (c) Downstream tasks

Figure 3: **Supervised: The optimal layer to cut might change depending of the training optimization, the data and the downstream task.** The best accuracy for each curve is show as a big square. For each experiments, we trained a headed supervised Resnet50 over ImageNet (with a 3 layer MLP as projection head). For a) and c) we trained this network over the full training set whereas for b) we use a random subset of 250 classes. Then, we froze the model parameters and trained linear probes over representation at different layers. **a)** We trained two models with different optimization pipeline: the first one in blue was trained with SGD using a cycling learning rate, along with momentum and weight decay. The second one in gray was trained with AdamW without additional regularization. This model is overfitting on the training set, which leads to similar validation performances across the backbone and projector. In contrast, the first one generalize much better but the performances across layers change significantly. **b)** Validation accuracy given by linear probes on different random subset of 250 ImageNet's classes for each layers. The validation split in gray corresponds to the same subset of classes that was used for training whereas Split 1-6 corresponds to different OOD random split. In this instance, we see that the optimal layer to use is the first layer of the projector. **c)** Validation performances on different downstream tasks. We have used the well regularized model from a) and evaluate it across different downstream tasks. For some datasets, the optimal layer to use is the last one, while for some other the optimal layer is the second layer of the projector.

**Misalignment between the training and downstream tasks while using different data distributions.** When using a pretrained model to predict new classes, there is a bias in the data distribution as well as in the fine-tuning objective (with respect to the training settings). We did a first experiment in Figure 3c in which we train a supervised Resnet50 over ImageNet. Then, we freeze the weights of the model and train a linear probe over ImageNet (Deng et al., 2009), CIFAR10 (Krizhevsky, 2009), Place205 (Zhou et al., 2014), CLEVR (Johnson et al., 2017) and Eurosat (Helber et al., 2019) at different layers. We observe that the readout performances on ImageNet are the best at the last layer but for datasets like CLEVR or Place205 the best performances are obtained at the second projector layer. In Figure 4, we performed the same experiment but this time using SimCLR. In this instance, the best performances for ImageNet are obtained at the backbone whereas the best performances for Eurosat, CLEVR and Imagenet 1% are obtained at the first projector layer. **This result challenges the common practice of discarding the entire projector in SSL since the layers to cut depend on the downstream task.**

**Misalignment between the training input data distribution and testing input data distribution while using the same training and downstream tasks.** Another type of bias can arise when using a wrongful data distribution after training of the model. This scenario is often referred to Out Of Distribution (OOD) since the distribution of the data used by the model becomes different from the one seen during training. We took the supervised model trained on ImageNet along with the linear probe trained at different layers and evaluate the performances of these readouts on ImageNet-C (Hendrycks & Dietterich, 2019) which is a modified version of the validation set of ImageNet on which different data transformations were applied. Our experiment in Table 1 demonstrates that the

| Head 3 | Head 2 | Head 1 | Trunk |
|--------|--------|--------|-------|
| 59.0 | 58.8 | **58.0** | 63.3 |

Table 1: ImageNet-C mCE (unnormalized) across layers.

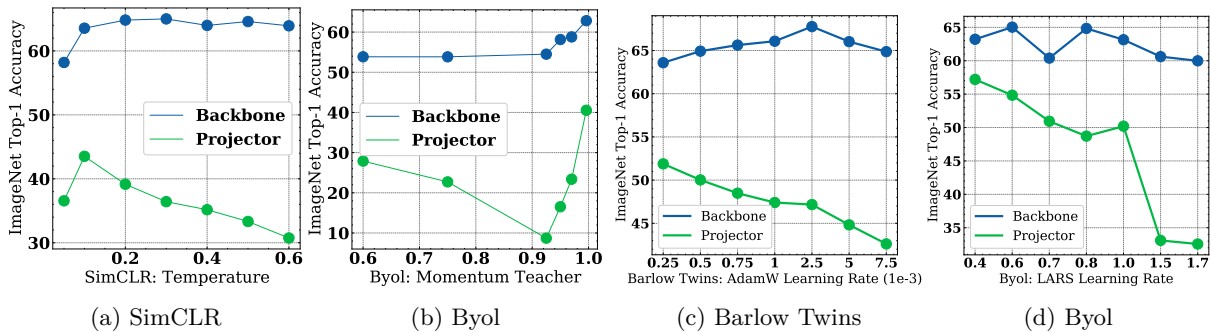

Figure 5: **The performances at the projector level aren't always correlated with the performances at the backbone level**. We train SimCLR, Barlow Twins and Byol with different hyper-parameters and evaluate with a linear prob, the performances at the backbone but also at the projector level on ImageNet classification task. For each model, we observe that the accuracy given by the linear probe at the backbone level isn't always correlated with the performance at the projector level.

performances are better after cutting two layers from the head of the network which highlight that it might be a good practice to probe intermediate representations when evaluating on OOD tasks.

### 3.3 The readout performances at the projector and backbone level are not always correlated

In Figure 5 we study the effect of Guillotine Regularization with respect to an hyper-parameter grid search for various SSL methods (SimCLR, Barlow Twins and Byol). When looking at the performances on ImageNet using a linear probe at the backbone level, one can observe an almost stable classification task performance for different hyper-parameters such as SimCLR temperature, Barlow Twins and Byol learning rate while the corresponding performances at the projector level change significantly. This highlights that the performances at the projector level are not always correlated with the performances at the backbone level. In consequence, knowing the performances of a linear probe at the projector level cannot give in advance insights about the performances at the backbone level.

## 4 Reducing the Need for a Projector in Self-Supervised Learning by increasing the alignment with the downstream task

Self-Supervised Learning is often considered a distinct learning paradigm in between supervised and unsupervised learning. In reality, the distinction is not as sharp, and much of SSL can be understood as solving a pretext-tasks akin to a supervised task(Wu et al., 2018; Khosla et al., 2020), merely with pseudo-labels obtained in another way than by human annotation. In this section, we show that different data selection process in SSL influences the alignment between the downstream and pretext task, which heavily impact the need of using a projector head in SSL.

To confirm the hypotheses that SSL methods need to use a projector because of a misalignment between the pretext and downstream task, we have to verify that reducing this misalignment, results in reducing the performance gap between the Trunk and Head representations. Ideally, we would like to get close to the supervised scenario in Figure 1 for which the optimal readout function is obtained at the last layer. To do so, we devise two experimental setups in which we replace the traditional data augmentation pipeline used in SSL, which consists of using handcrafted augmentation on each image to create a set of pairwise positive samples.

**In the first setup, while using the exact same SSL criterion (SimCLR), we use as positive examples pairs of images that belong to the same class, and as negative examples images that don't belong to the same class.** Note that the SSL training criteria will push towards a collapse in the representation space of all the images belonging to the same class, while pushing further apart the different

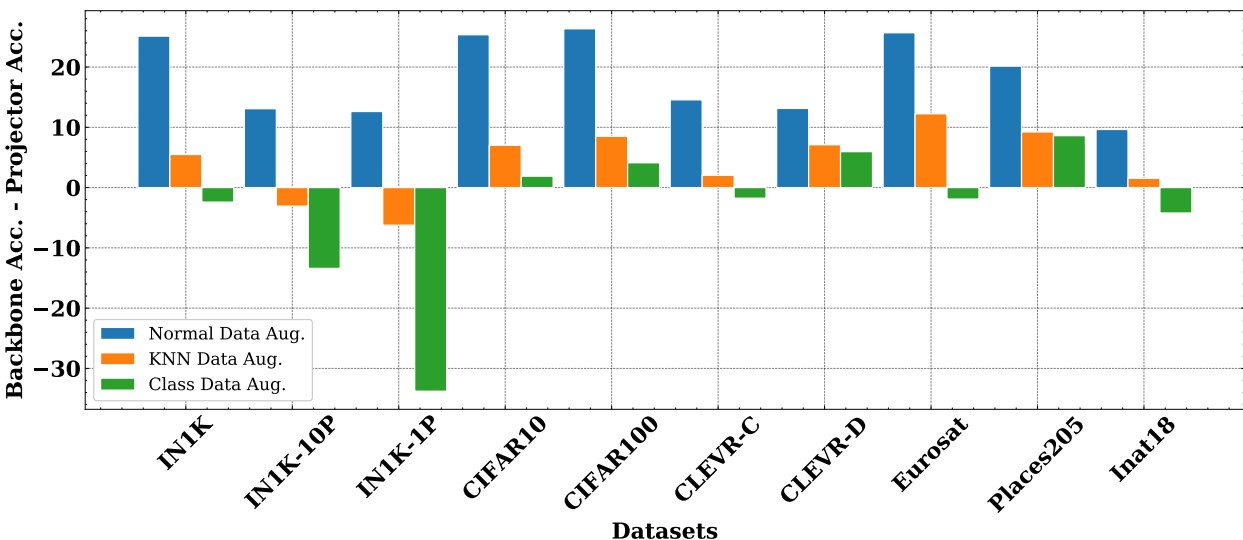

Figure 6: Difference in accuracy with linear probing between the projector and backbone representation with different alignments with respect to the classification downstream task. In this experiment we used SimCLR and we change how the positive pair are defined to better aligned with a classification downstream task. In blue, our baseline, we trained SimCLR with the traditional SSL data augmentations which defines the positive view as two augmentations of a same image. In orange, we use the embedding of a pretrained model to define the positive pair as two nearest neighbors under a pretrained model (while using the same data augmentation as the baseline). In green, we use a supervised class label selection to define the positive examples. In this scenario, SimCLR should learn to produces similar embedding to all images belonging to a given class. All three models are trained on ImageNet (IN1K), then we evaluate them with a linear probe across a wide range of downstream tasks at the backbone and projector level and show the difference in accuracy between both. **When the difference is positive, the accuracy at the backbone level is higher than the one at the projector level, highlighting the benefits of Guillotine Regularization. In contrast when the difference is negative, the accuracy at the projector level is higher than the one at the backbone level. In this instance, Guillotine Regularization is not needed.** When positives pairs are defined as belonging to a given class, there is no misalignment with the imagenet classification downstream task. Thus on ImageNet-1K, ImageNet1k-10P (10% of the training set to train the linear probe) and ImageNet1k-1P (1% of the training set to train the linear probe), we observe that the performances at the projector level are much higher than the ones at the backbone level. Interestingly, the nearest neighbors heuristic reduces considerably the impact of Guillotine Regularization across several downstream tasks.

class clusters. By doing so the training SSL objective becomes perfectly aligned with the downstream classification task, despite using a SSL training criteria instead of a traditional cross entropy loss.

**In the second setup, we use as positive pairs the closest neighbors found by a pretrained SSL model trained with the traditional SSL handcrafted data augmentation pipeline.** The reasoning is that if instead of considering each image of the dataset as its own specific class, we use clusters of many images to define the positive pairs, we might be able to close the gap with respect to a supervised baseline without the need of labels.

In Figure 6, we show the differences in accuracy between the backbone and the projector with respect to these two new data augmentation scenarios. The baseline, using the traditional SimCLR positive pairs based on data augmentations is in blue, the nearest neighbors setup in orange and the class based setup in green. We observe for SimCLR that using the nearest neighbors based heuristic is helping in reducing the gap between the pretext and downstream task while having a purely supervised heuristic to define the positive pair is removing the need to perform Guillotine Regularization across several downstream tasks. Hence confirming the hypothesis that the effectiveness of a projector depends of the alignment between the pretext and downstream task in self-supervised learning.

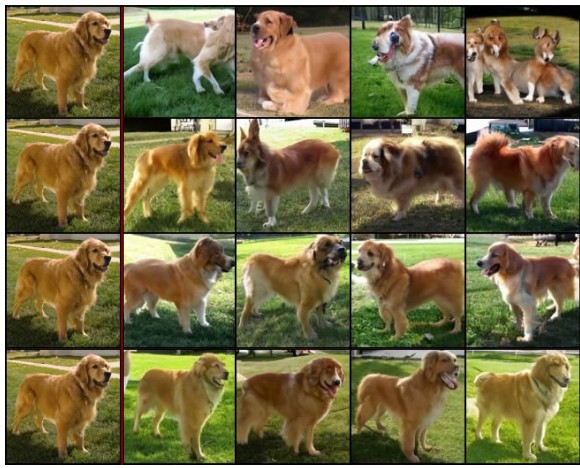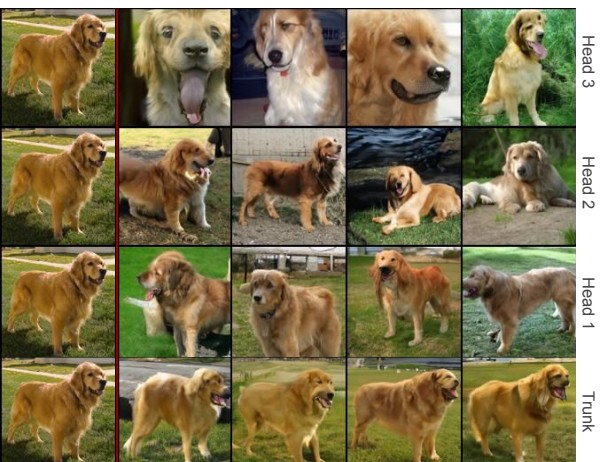

(a) SimCLR trained with SSL augmentations.     (b) SimCLR trained with class labels.

Figure 7: In this figure, we used RCDM (Bordes et al., 2022), a conditional generative model to visualize what information is decodable at different layers. The leftmost column of images (before the red line) is the conditioning image that was used to compute the representation that is fed to RCDM. The subsequent columns are samples generated by the model using this representation. The first row correspond to the last projector layer, the second row to the second projector layer, the third one to the first projector layer and the last row to the backbone layer. As show in Figure 6, changing the alignment with the pretext task change significantly the information encoded by the neural network. When using SSL augmentations at the projector level, the information about the dog's breed seem to have been lost whereas when looking at a network trained with supervised augmentations, the information is preserved throughout each layers.

## 4.1 Visualizing the information across layers for different alignments

In this section, we use RCDM (Bordes et al., 2022), a conditional generative model to visualize what information is retain or not in the representation. We train RCDM on ImageNet with blurred faces(Yang et al.), using the representation given by a SimCLR model trained on handcrafted SSL views and another which was trained on class based views. In Figure 7, we show that when looking at different decoding corresponding to different layers in the network, the information encoded vary a lot depending on the layer to use. When going deeper, RCDM is not able to reconstruct as much as information about the images than when using the backbone representation (which contain much more low level features). When looking at the generated samples that were conditioned on the representation of the model trained with supervised views, we observe that the breed of the dog stay the same across layers. However when using traditional data augmentations, the information about the specific golden retriever breed is lost in the last projector layers. This is correlated with the fact that this model get lower classification performances when using the projector.

## 4.2 Experimental details

We use Pytorch (Paszke et al., 2019) and FFCV-SSL (Bordes et al., 2023; Leclerc et al., 2022) as data loader. All the experiments were performed with a Resnet50 (He et al., 2016) (except if mentioned otherwise) as backbone. For each model, we use a batch of size 2048 and AdamW (Loshchilov & Hutter, 2019) as optimizer with an adaptive learning rate schedule. We run the training for 100 epochs. For each model, we add as head a small MLP of 3 layers of size 2048 (same dimension as the backbone) with ReLU (Glorot et al., 2011) as activation and batch normalization (Ioffe & Szegedy, 2015). When training different SSL methods, we always used the same set of data augmentations (with cropping, color-jitter, random grayscale, gaussian blur and solarization).

## 5 Conclusion

Through empirical evaluations, we demonstrated that the optimal layer to use for downstream evaluation vary depending on several factors: optimization, data and downstream task. These results highlight the need for SSL practitioners to run systematic evaluations at several layers instead of using always the backbone as reference. We also demonstrated that the use of a projector in SSL depends on the alignment between the downstream and pretext task. Despite, its usefulness, having to rely on a *trick* like Guillotine Regularization to increase performances reveals an important shortcoming of current self-supervised learning methods: the inability to design experimental setups and training criteria that learn structured and truly invariant representations with respect to an appropriate set of factors of variation. As future work, in order to escape from Guillotine Regularization, we should focus on finding new training criteria and data augmentations that will be more *aligned* with the downstream tasks of interest.

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

## A  Datasets

In this work, we use ImageNet (Deng et al., 2009) (Term of license on https://www.image-net.org/download.php) for our experiments. We also used a synthetic 3D dataset that will be described in the next subsection.

### A.1  3D models dataset

We will now discuss the dataset used for figure 2. As previously mentioned, this dataset consists of 3D models from 3D Warehouse (Trimble Inc), freely available under a General Model License, and rendered with Blender's Python API. We alter the scene by uniformly varying the latent variables described in table 2.

The variety in the scenes that can be generated is illustrated in figure 8. We can see that each latent variables can significantly impact the scene, giving a significant variety in the rendered images.

## B  Reproducibility

Our work does not introduce a novel algorithm nor a significant modification over already existing algorithm. Thus, to reproduce our results, one can simply use the public github repository of the following models: SimCLR, Barlow Twins, VicReg or the PyTorch Imagenet example (for supervised learning) with the following twist: adding a linear probe at each layer of the projector (and backbone) when evaluating the

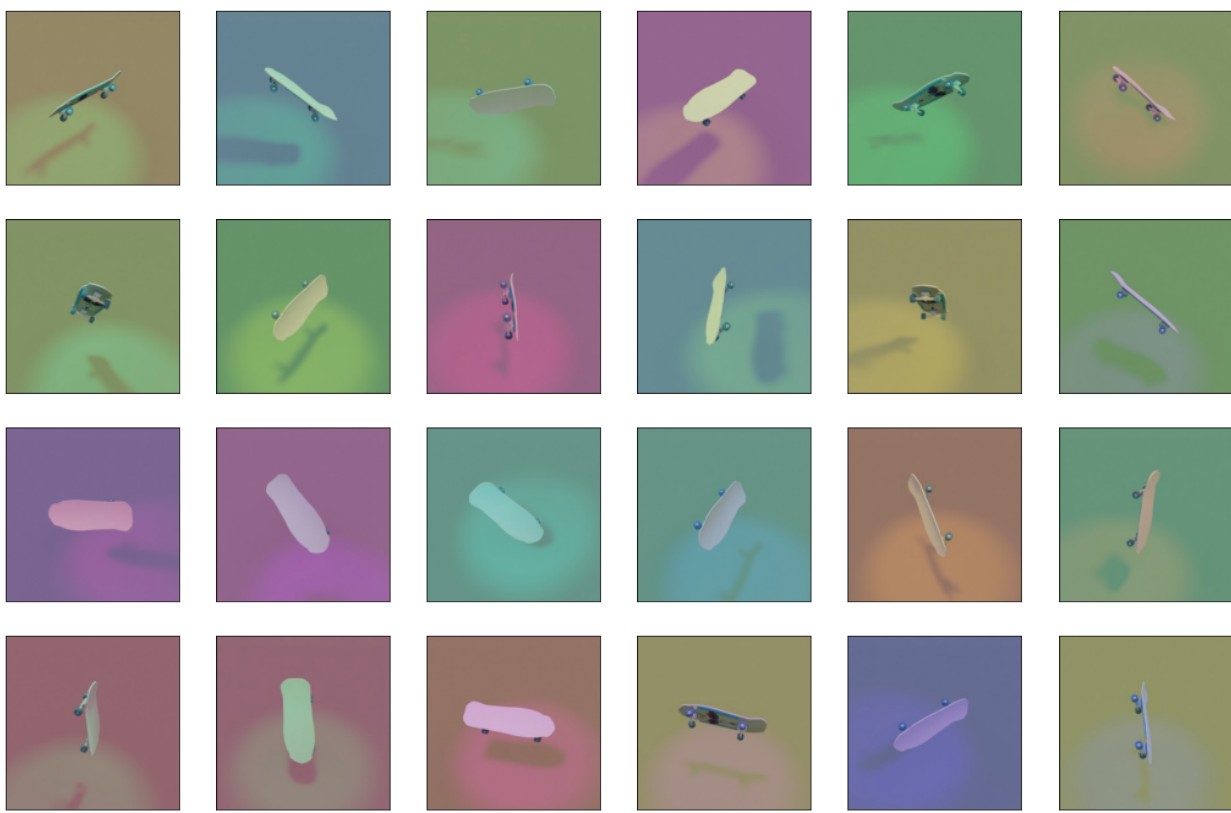

Figure 8: Rendered views of a skateboard generated by randomly sampling latent variables. The influence of each parameter is easily visible, which is expected to make their prediction easier.

Table 2: Latent variables used to generate views of 3D objects. All variables are sampled from a uniform distribution.

| Latent variable | Min. value | Max. value |
|---|---|---|
| Object yaw | $-\pi/2$ | $\pi/2$ |
| Object pitch | $-\pi/2$ | $\pi/2$ |
| Object roll | $-\pi/2$ | $\pi/2$ |
| Floor hue | 0 | 1 |
| Spot $\theta$ | 0 | $\pi/4$ |
| Spot $\phi$ | 0 | $2\pi$ |
| Spot hue | 0 | 1 |

model. However, since many of these models can have different hyper-parameters, or data-augmentations, especially for the SSL models, we recommend to use a single code base with a given optimizer, a given set of data augmentations so that comparisons between models are fair and focus on the effect of Guillotine Regularization. In this paper, except if mentioned otherwise, we use as Head, a MLP with 3 layers of dimensions 2048 each (which match the number of dimensions at the trunk of a Resnet50) along with batch normalizaton and ReLU activations.

## C   Additional experimental results

In this section, we present additional experimental results. The first one in Figure 9 is an extended version of Figure 1 with additional results on the training set. Figure 10 is a similar setup to the one in Figure 9

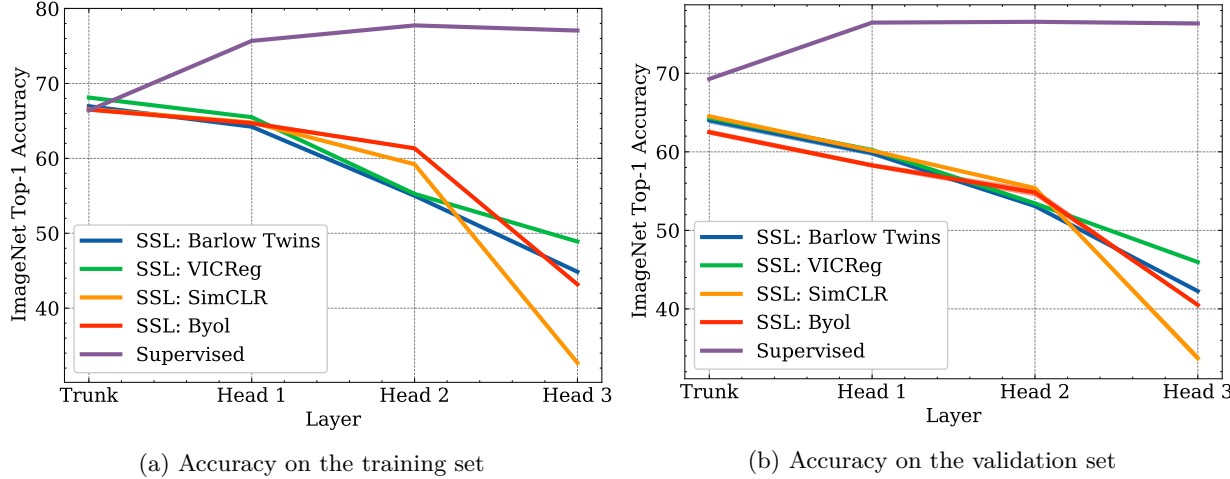

(a) Accuracy on the training set

(b) Accuracy on the validation set

Figure 9: We measure with linear probes the accuracy at different layers on a resnet50 (as Trunk) on which we added a small 3 layers MLP (as Head) for various supervised and self-supervised methods on the training and validation set. For each method, we show the mean and standard deviation across 3 runs (The std between different runs is low). When looking at self-supervised methods, the gap in performances between the linear probe trained at different levels can be as high as 30 points of percentage.

where we compared the performances at different layers for SSL methods and a supervised one except that we use a VIT-B instead of a Resnet50. We observe an important gap on the classification performances reached with a linear probe on different layers with the VIT-B when using SSL methods.

In Figure 11, we show how the performances at different layers change during training by using an online linear probing. At the beginning of the training the gap of performances between layers is low, however it increases significantly after 10 epochs.

In Figure 12 we show the accuracy computed with linear probes trained using projector and backbone representations. This figure is similar to Figure 6 except that we present the absolute accuracy value instead of the difference in accuracy with respect to the backbone.

## D   Limitations

In this work we focused mostly on analyzing the use of Guillotine Regularization in the context of Self-Supervised Learning. However, this kind of regularization might be useful for a variety of other types of training methods which we don't investigate in this paper. We also mostly focus on generalization for classification tasks, but other tasks could also been worth exploring.

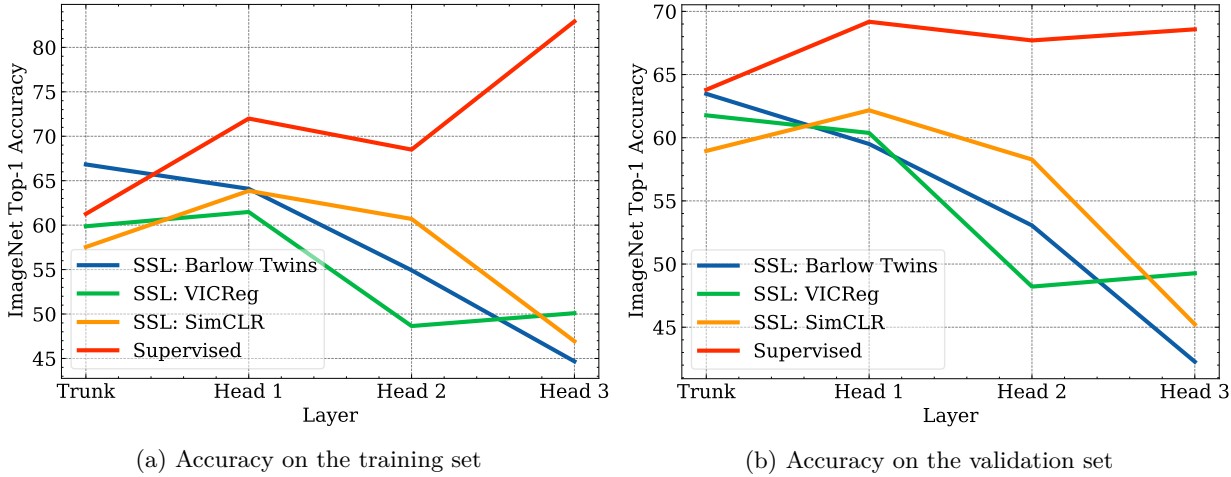

(a) Accuracy on the training set

(b) Accuracy on the validation set

Figure 10: Same experiment as in Figure 9 but this time, we measure with linear probes the accuracy at different layers on a VIT-B (as Trunk) on which we added a small 3 layers MLP (as Head) for various supervised and self-supervised methods. Since the outputs of the VIT-B has a lower number of dimensions than a Resnet, we added at the trunk of the VIT-B a linear layer with ReLU activation to project into a 2048 dimensional vector. In the supervised learning setting, the best performances are obtained when using the last layers of the model. But, when looking at self-supervised methods, the gap in performances between the linear probe trained at different levels can be as high as 20 points of percentage. Interesting, it seems for the VIT-B that we got the best performances at Head 1 for SimCLR whereas for the ResNet, the best performances were obtained at the Trunk. It is likely that for different architectures, the optimal number of layers on which to apply Guillotine Regularization will vary.

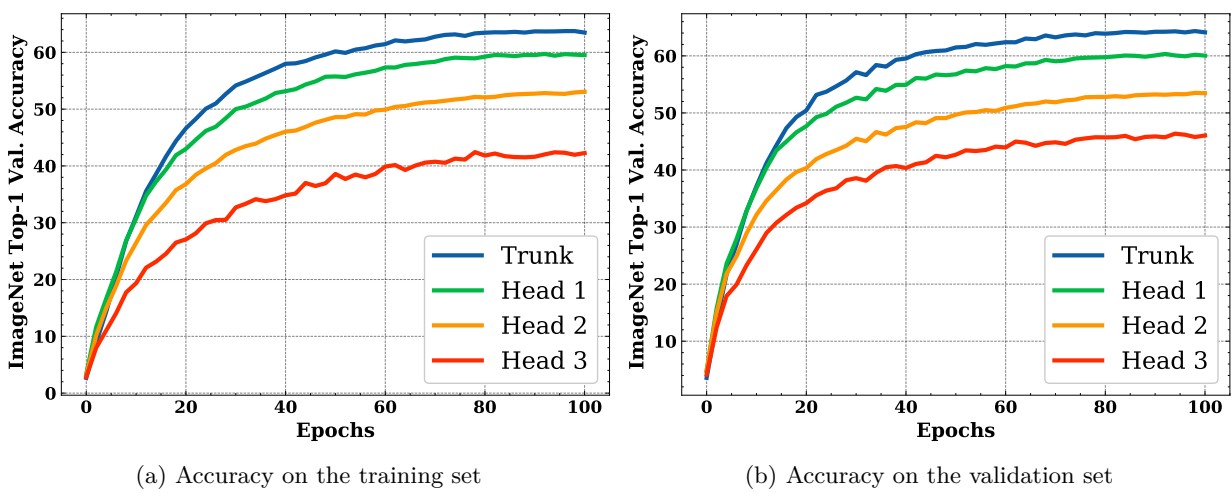

(a) Accuracy on the training set

(b) Accuracy on the validation set

Figure 11: a) Accuracy of Barlow Twins through epochs computed with online linear probing at different layers. At the beginning of the training the gap in performances between the probes is small however after 10 epochs, the gap becomes larger and larger both on the training and validation set.

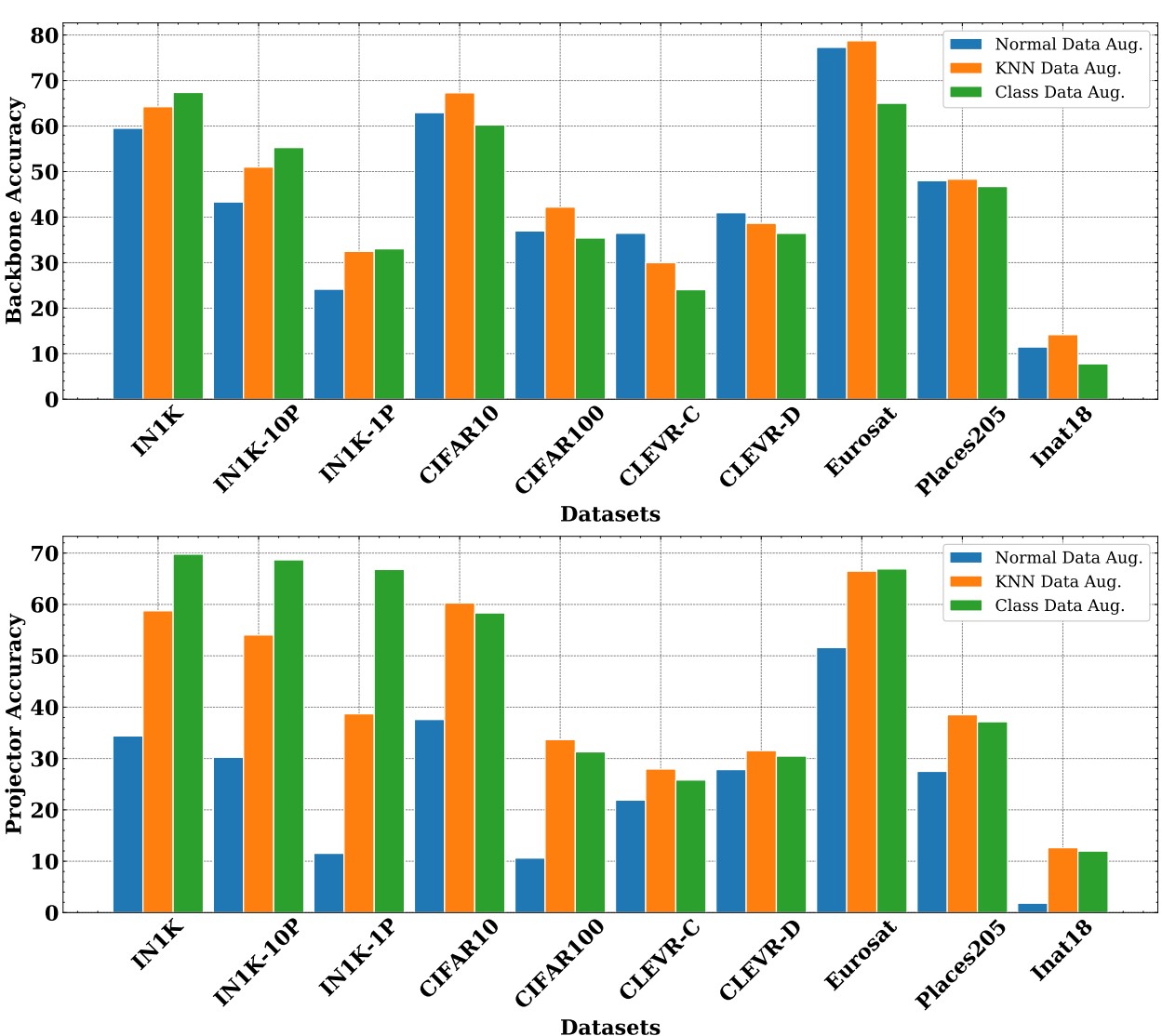

Figure 12: **Backbone and projector accuracy with linear probing with different alignment with respect to the classification downstream task.** In this experiment we used SimCLR and we change how the positive pair are defined to better aligned with a classification downstream task. In blue, our baseline, we trained SimCLR with the traditional SSL data augmentations which defines the positive view as two augmentations of a same image. In orange, we use the embedding of a pretrained model to define the positive pair as two nearest neighbor under this pretrained model (while using the same data augmentation as the baseline). In green, we use a supervised class label selection to define the positive example. In this scenario, SimCLR should learn to produces similar embedding to all images belonging to a given class. All three models are trained on ImageNet (IN1K), then we evaluate them with a linear probe across a wide range of downstream tasks at the projector and backbone level. When positives pairs are defined as belonging to a given class, there is no misalignment with the imagenet classification downstream task. Thus on ImageNet-1K, ImageNet1k-10P (10% of the training set to train the linear probe) and ImageNet1k-1P (1% of the training set to train the linear probe), we observe that the performances at the projector level are much higher than the ones using the traditional SSL augmentations. Interestingly, the nearest neighbors heuristic reduces considerably the impact of Guillotine Regularization across several downstream tasks.

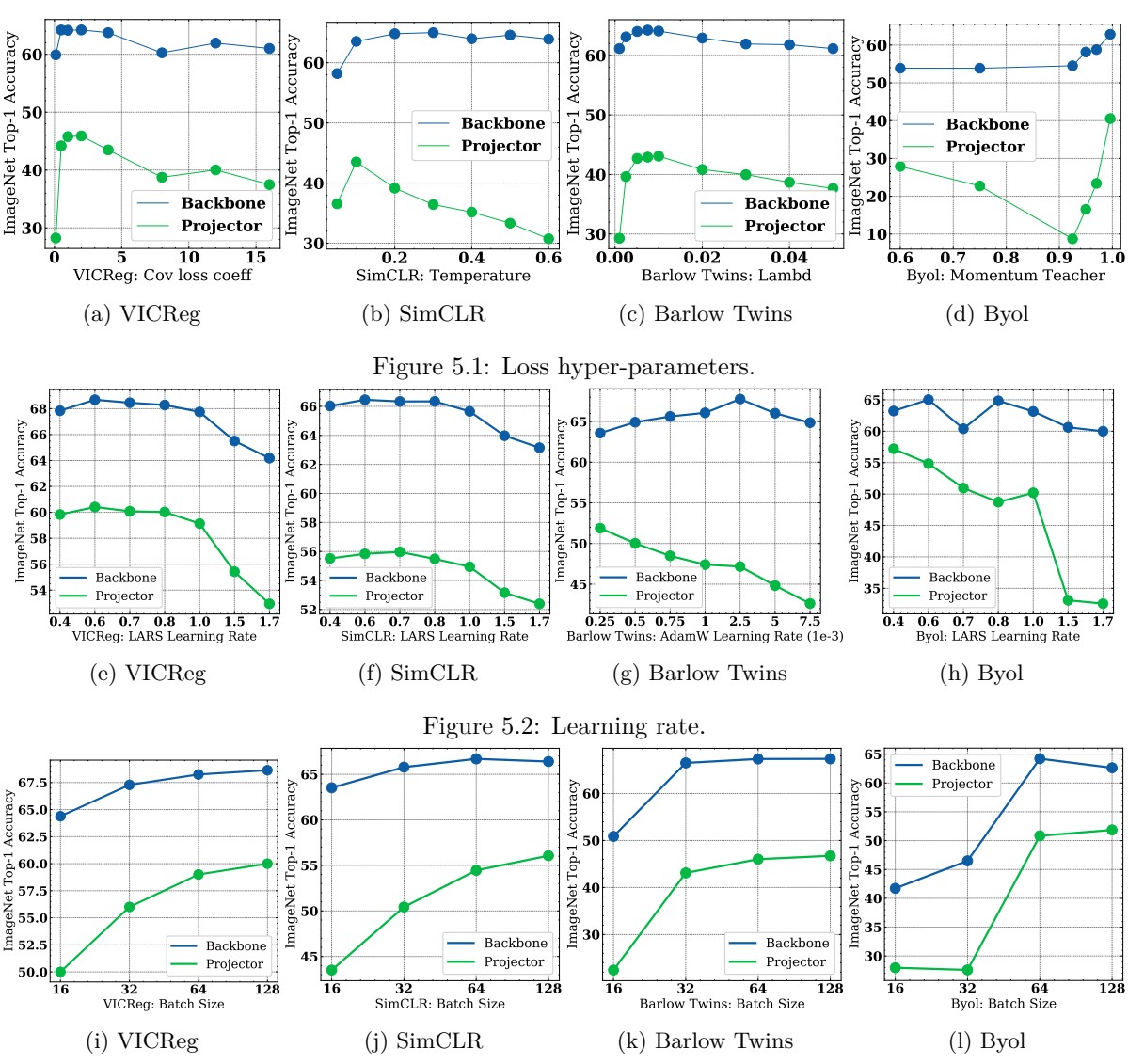

Figure 13: How different hyper-parameters impact the gap in performances between the backbone and projector representation. We train SimCLR, VicReg, Barlow Twins and Byol with different hyper-parameters and evaluate with a linear prob, the performances at the backbone but also at the projector level on ImageNet classification task. For each model, we observe that the accuracy given by the linear probe at the backbone level is fairly stable across the grid search of hyper-parameters while the linear probe at the projector level can reach very low accuracy. This highly that the probes performances at different layers might not be always correlated with each others.

