# OpenReview forum: "Guillotine Regularization: Why removing layers is needed to improve generalization in Self-Supervised Learning"
_TMLR — Accepted by TMLR_

### Review · Reviewer_mA1b · 2023-02-05

**Summary Of Contributions:**

This paper formalizes Guillotine Regularization (GR), a regularization technique which involves removing the last head of a pre-trained neural network to improve generalization in downstream tasks for self-supervised learning (SSL). They demonstrate that GR greatly improves performance in situations where the (pre-) training procedure is misaligned with the downstream task on Imagenet, a variety of transfer learning tasks (e.g. ImageNet -> Place205), and a synthetic 3D dataset.

**Audience:**

Yes

**Broader Impact Concerns:**

There are no immediate ethical concerns or broader impact considerations that are evident in this submission.

**Claims And Evidence:**

Yes

**Requested Changes:**

I don’t think the paper necessarily requires a theoretical explanation for why GR works well in practice. So I believe it’s fair for the submission to rephrase claims of providing theoretical justifications of the method to something more along the lines of providing conjectures. However, if the authors do actually want to corroborate their conjectures, it would be interesting to perform experiments similar to (Tian et al. 2020) where they compute neural estimates of mutual information (MI) between H_L, H_{l+1}, y_source, y_target, etc. to assess whether removing more layers in the head (e.g. representation H_L) indeed has higher MI with y_target than H_{l+1}, etc.

I also think that the next version of this paper should include missing citations, such as (Wu et al. 2018), (Tian et al. 2020), and (Khosla et al. 2021). There are also a variety of mutual information-inspired SSL methods such as (Tschannen et al. 2019), (Wu et al. 2020), and (Taghanaki et al. 2021).


**Strengths And Weaknesses:**

Strengths:
- The experiments were interesting – especially the empirical benefits of GR across various datasets and head layers removed – and would be a good contribution. I liked the experiments shown in Figure 3, as well as the experiments in Section 4 that are trying to assess the impact of source/target task mismatch with the particular choice of the pretext task (positive pairs are from the same class) (Khosla et al. 2021).

Weaknesses:
- The paper claims to provide a theoretical motivation (Sections 3.1 - 3.2) as to why GR works so well in practice – this part of the paper is very limited. In particular, the analysis is a handwave-y explanation of information bottleneck (and the data processing inequality). However, it wasn’t clear to me how the information bottleneck formulation relates to the relationship between the source task and the target task. This part seems critical. Is GR saying that “processing the data less” (removing more layers of the head) leads to less compression of X via H_L, and that this is beneficial because there is more information about Y_target potentially stored in H_L rather than H_{l+1}? But if this were true, wouldn’t a somewhat silly argument here be that GR should work better as we remove more layers in the head? I think this is what the authors may have been trying to get at with the 3D experiments (with the observation that removing more layers is beneficial for improving performance as the gap between the source and target tasks increase), but it’s very hard to reason about Y_source and Y_target in this way.

- There is also an observation of how certain SSL objectives such as SimCLR reduce to the traditional cross entropy loss in the full batch setting (Section 4.1). This is also not new, and the observation that SSL can also become an “image index classification task” is already known (Wu et al. 2018). There are also several existing works looking at pretext tasks that look at class membership when considering data augmentations, such as (Khosla et al. 2021).

---

> ### Author Response · Authors · 2023-03-02
> **Answer to Reviewer mA1b**
>
> Thank you for your review. We invite you to read the general answer to reviewers as well as the updated pdf.
>
> > The paper claims to provide a theoretical motivation (Sections 3.1 - 3.2) as to why GR works so well in practice – this part of the paper is very limited ... it’s very hard to reason about Y_source and Y_target in this way.
>
> We agree that this part was confusing and misleading. We removed it since it wasn't the main focus of the paper. However, we still provide an high level intuition in Figure 2 that don't assume anything about the flow of information going through the network. We hope that this new justification is more clear than the previous one.
>
> > There is also an observation of how certain SSL objectives such as SimCLR reduce to the traditional cross entropy loss in the full batch setting ... considering data augmentations, such as (Khosla et al. 2021).
>
> It's true that the link between SSL objectives and traditional cross entropy loss was already made in the literature. In consequence, we removed section 4.1 and focus section 4 around reducing the misalignement between the pretext and downstream task.
>
> > I also think that the next version of this paper should include missing citations, such as (Wu et al. 2018), (Tian et al. 2020), and (Khosla et al. 2021). There are also a variety of mutual information-inspired SSL methods such as (Tschannen et al. 2019), (Wu et al. 2020), and (Taghanaki et al. 2021).
>
> We included the missing references except the ones around mutual information since we remove the analysis around MI in the paper.
>
> Please let us know if there is anything else and thanks again for your time.

---

### Review · Reviewer_JDve · 2023-02-08

**Summary Of Contributions:**

This paper delves into the circumstances under which the removal of the final layers of a pre-trained network, named Guillotine Regularization (GR), can improve performance. Through experimentation on various datasets, the authors argue that GR yields the best results when the target and source tasks are significantly different. GR also seems to reduce fluctuations in performance resulting from untuned hyperparameters.

**Audience:**

Yes

**Broader Impact Concerns:**

I see no clear broader impact concerns stemming from this work.

**Claims And Evidence:**

Yes

**Requested Changes:**

I would appreciate if the authors could try to alleviate the weaknesses I listed before by trying to delve deeper in their main insights with different techniques.

I also feel that toning down the message regarding importance of hyperparameter-tuning with GR is important to improve the factuality of this work.

Finally, I believe this paper could also mention/discuss prior work on layer specialization from the noisy label literature, and the example difficulty literature, e.g., (Maennel et al. 2020) and (Baldock et al. 2021).

- Maennel et al. What Do Neural Networks Learn When Trained With Random Labels? NeurIPS 2020
- Baldock et al. Deep Learning Through the Lens of Example Difficulty. NeurIPS 2021

**Strengths And Weaknesses:**

## Strengths

1. **Clean writing**: This paper is very well-written and fun to read.
2. **Correctly executed experiments**: The experiments are well-designed and executed. They mostly support the discussed findings.
3. **A few novel results**: As far as I know, this particular set of experiments has not been conducted before, and the discussed findings regarding the specialization ability of the last layers of SSL pretrained networks are probably new.

## Weaknesses

1. **Little added value over prior work**: In my opinion, the main weakness of this work is the fact that the discussed findings are a direct translation of those already known for supervised learning to SSL. In this regard, the proposed view of SSL as a specific subinstance of supervised learning is not new, and therefore, one could have probably expected the presented results even without having performed these experiments. In fact, some of the presented experiments (e.g., training and evaluation on different subsets of ImageNet) are copied almost verbatim from prior work (Yosinski et al., 2014), which further adds to the feeling that the contributions of this work are very small. The evaluation of these experiments on SSL are still technically new, however, so this might be enough to meet the bar for acceptance to TMLR.
2. **Insights on effect of GR in hyperparameter variability are a bit subjective**: Personally, looking at the plots in Fig. 5.3., I fail to agree with the authors that GR ensures "a fairly stable accuracy across hyperparameters". In my opinion, there is indeed a smaller difference in performance, but there are still big fluctuations with respect to different hyperparameter values. The way the authors phrase this section it sounds as though GR allows for no hyperparameter tuning, and in light of these results this is clearly not true.
3. **Shallow findings**: Considering that most findings could have been predicted in hindsight, and that most experiments are not very novel, I would have appreciated if the paper had studied this problem in more depth, and provided further intuitions. This could be done by visualizing feature maps/embeddings, studying GR in light of the results in Evci et al. (2022), or describing the effectiveness of GR at different pretraining stages, for example.

---

> ### Author Response · Authors · 2023-03-02
> **Answer to Reviewer JDve**
>
> Thank you for your review. We are glad that you found the paper pleasant to read. We invite you to read the general answer to reviewers as well as the updated pdf.
>
> > Little added value over prior work: In my opinion, the main weakness of this work is the ... these experiments on SSL are still technically new, however, so this might be enough to meet the bar for acceptance to TMLR.
>
> It’s true that these results were known for a long time in the supervised learning literature. We don't claim any novelty but we consider our work as a bridge between these previous results and the SSL literature. Mostly because there has been a lot of misconceptions in the SSL literature around the fact of throwing away the projector and why it’s needed in the first place. Our main contribution is to show that intermediate projector layers can also give good performances on downstream tasks. In addition  several works Wang et al. (2022),Sarıyıldız et al. (2023) have shown that using a projector in a supervised network helps generalization. When viewing the projector trick as something similar to what was done in transfer learning, those results are not surprising anymore. In consequence, we think that our work is important for the community to realize there isn't anything specific or new related to the use of throwing away a projector. Otherwise, more people will continue to “remove” the projector in SSL without having in mind the transfer learning literature and more people will discover again that having a projector helps in a supervised setting without considering that cutting layer is a common technique. By giving a single *fun* name to this technique, we hope that researchers in SSL will realize how common the technique is and that rigorous readout evaluations at different layers of representation is important. In consequence, we still believe that the connections we made along the empirical evidences we provide will be of interest to at least some individuals in TMLR's audience.
>
> > Insights on effect of GR in hyperparameter variability are a bit subjective ...
>
> We agree that this part of the paper was not clear. In this plot, we mostly wanted to show that the accuracy at the projector level might not be correlated with the performance at the backbone level. In some of these plot, a 20% or 50% accuracy at the projector level could give the same accuracy at the backbone level. In such instances, the backbone representation seems to be robust to different solutions/type of representation at the projector level; however, as you said, this robustness is not absolute and there are instances when the backbone and projector representation give bad performances. So the main conclusion of the plot was that the performances at the backbone aren’t always correlated with the performances at the projector. We try to make this idea more clear in the last revision of the paper.
>
> > Shallow findings: Considering that most findings could have been predicted in hindsight, and that most experiments are not very novel, I would have appreciated if the paper had studied this problem in more depth, and provided further intuitions. This could be done by visualizing feature maps/embeddings, studying GR in light of the results in Evci et al. (2022), or describing the effectiveness of GR at different pretraining stages, for example.
>
> Since we removed the part around the theoretical motivation, we increased the number of experiments to provide more insights. We used RCDM to visualize the feature maps/embeddings at different layers in Figure 8 for different SimCLR pretraining strategies. We also add the study of the effect of GR across different pre training stages in Figure 12. We also added in Figure 4, a) an example of how different training setup influences the intermediate layer performances. Please let us know if there are any additional experiments you would like to see.
>
> > I also feel that toning down the message regarding importance of hyperparameter-tuning with GR is important to improve the factuality of this work.
>
> We toned down the hyperparameter-tuning message in the paper.
>
> > Finally, I believe this paper could also mention/discuss prior work on layer specialization from the noisy label literature, and the example difficulty literature, e.g., (Maennel et al. 2020) and (Baldock et al. 2021).
>
> Thank you so much for the missing references, we added them in our last revision.

---

> > ### Comment · Reviewer_JDve · 2023-03-13
> > **Reply to authors**
> >
> > Thank you very much for your answer and for taking the time to revise the paper based on the feedback of all the reviewers. Personally, I believe this new version is better positioned with respect to the prior literature and has a much more precise and useful message. In particular, although I think the contribution of this work is small as most findings where already discussed in the supervised literature, I should agree with the authors that highlighting these results in the context of SSL is a valid and useful contribution. Nonetheless, there are a few points in the revised version that should be improved in my opinion:
> >
> > - **Figure. 2**: As it stands right now, Fig. 2. is really hard for me to understand. I fully understand the caption and the message that the authors are trying to give, but I do not intuitively see the meaning of the illustrations. I would recommend a complete revision of this plot, or even removing it altogether. should this manuscript be published.
> > - **Incorrect reading of Table 1**: As far as I know, mCE is an error metric, and as such the lower the mCE the more robust a model is. However, the comments of the authors in this regard say the opposite, and therefore lead to the wrong conclusions. Given the results in Table 1, the most robust representation cannot be found at the trunk, but at the heads. I therefore urge the authors to remove these results from the paper and any comments about the role of Guillotine Regularization in robustness to distribution shifts.
> > - **Too general title**: Since the main contributions of this work are limited to the SSL framework, I would encourage the authors to rethink the title of their paper to make this explicit rather than suggesting their work gives new general guidelines for deep learning.

---

> > > ### Author Response · Authors · 2023-03-15
> > > **Reply to Reviewer JDve**
> > >
> > > Thank you very much for your reply. We appreciate that the new version of the paper answered your concerns.
> > >
> > > - We updated a new version of the pdf in which we removed Figure 2 as requested. We agree that this figure didn't provide any additional information that wasn't already present in the text.
> > >
> > > - Thank you for pointing this out ! We updated the corresponding paragraph but we don't think it changes the conclusion which is to say that, depending on the training, data or downstream task, the optimal layer to use to probe the representation might vary. In this instance, the best layer to use, the one with the smallest error, is after cutting two layers from the projector. So it still fit very well with the point we are trying to make: it's a good practice to empirically probe intermediate representation, even in an OOD setting. The conclusion of this experiment would have been wrong only if the lowest error was achieved at the last projector layer: Head 3 (since it's the optimal layer to use on plain validation ImageNet as noted in Figure 3, c) ) which is not the case here. However, we are open to remove this part if the reviewers think we don't provide enough empirical evidences that probing intermediate representation might also be needed in the OOD setting.
> > >
> > > - We updated the title with the following: Guillotine Regularization: Why removing layers is needed to improve generalization in Self-Supervised Learning.

---

> > > > ### Comment · Reviewer_JDve · 2023-03-24
> > > > **Thank for the updates**
> > > >
> > > > I thank the author for making the requested changes. Personally, I believe the paper is now more precise and possibly informative to some audience to TMLR. The significance of the results is a bit limited, but I believe the results shown are correct, the insights sound, and the paper provides some added value. According to the TMLR guidelines, I will weakly vote for acceptance of this paper.

---

### Review · Reviewer_hcdo · 2023-02-16

**Summary Of Contributions:**

The main contributions in this paper are a series of empirical studies of the properties of the projection head of self-supervised learning methods. Specifically:
- When varying several parameters when synthesizing 3D renderings of objects and then trying to recover the value of these parameters using a linear probe, the optimal layer varies.
- When training linear probes on classification datasets on different head layers of a supervised network trained with a projection head, the backbone is usually better than later layers.
- When training linear probes on different head layers on ImageNet and then evaluating on OOD datasets, the backbone is again better than the later projection head layers.
- When training networks with contrastive loss using different sets of positives, the backbone is much better than the projector when the positives are augmented views of the same image, but its advantage is not as clear when positives are nearest neighbors in the representation space of a pretrained SSL model, and when the positives are other members of the same class the projector is often better than the backbone.
- ImageNet linear eval accuracy measured from the backbone is more robust to choice of hyperparameters (especially temperature) than linear eval accuracy measured at the projection head.

**Audience:**

Yes

**Broader Impact Concerns:**

No concerns.

**Claims And Evidence:**

No

**Requested Changes:**

- Unless there is a way to fix the theoretical claims, they should be stated only as intuition or removed, and they should not be claimed as a contribution.
- The description of the experiments in Figure 3c should be made consistent. Ideally the paper would also report mCE for each layer in addition to performance on the individual corruptions.
- If the authors want to keep Section 4.1, it should cite previous work that has made similar observations, but in my opinion it does not contribute much to the paper and could be removed.
- The papers above should be cited.
- I think it's fine to use "Guillotine Regularization" as a catchy title but I'd suggest minimizing the use of the term elsewhere. It's unnecessary and potentially confusing to rename the use of a projection head given that it is already a common technique. The abstract should also explicitly refer to an "MLP projection head" or "MLP projector" since that is what people commonly call this trick in previous literature.

**Strengths And Weaknesses:**

Strengths:

- The question of why the projection head improves linear evaluation performance in the context of self-supervised learning is an interesting one and is not satisfactorily addressed in existing literature.
- The submission contains several interesting experiments looking at properties of different layers of networks trained with a projection head.
- The gains in OOD accuracy by using the backbone vs. projection head output in Figure 3c are particularly interesting to me, although as noted below, I am not sure I understand the details of the experiment.

Weaknesses:

- The intro lists as one of the paper's contributions "To formalize and motivate theoretically the common trick of using a projector in Self-Supervised Learning" but I do not believe it makes a meaningful contribution here. The "formalization" in Section 3.1 is just notation and the Information Theoretic Motivation on p. 5 is conceptually similar to the justification for the projection head provided in Section 4.2 of Chen et al. (2020).
- Since Chen et al. (2020)'s justification for the projection head is informal, the Information Theoretical Motivation could still be a contribution of interest to TMLR's audience if it were correct. However, the argument that neural networks lose information as information propagates from one layer to the next is not convincing, as previously noted by Goldfeld et al. (2019). For the case of continuous inputs, the mutual information between the input and the output of a deterministic neural network is usually infinite. To demonstrate this in a ReLU net, it suffices to show that there is at least one region of the input over which the entire network is linear. For the case of discrete inputs, in order to be information-preserving, the network need only be injective, not bijective. For networks with strictly monotonic activations, the network is injective except at a measure zero set of weights. In principle it is possible for ReLU networks to lose information, but that would require that there are two inputs that produce the same output.
- The paper seems to provide two contradictory descriptions of what is shown in Figure 3c. At the bottom of p. 6, the paper says that the linear probes are trained on the standard ImageNet classification task and evaluated on ImageNet-C. However, the caption of Figure 3 says the probes are trained on ImageNet-C.
- I wonder whether how robustness measured on ImageNet-C at the backbone of a network trained with a projection head compares with robustness of a standard ResNet-50 with no projection head and would like to compare the numbers with other work. For that purpose, it would be useful to report mCE on ImageNet-C as suggested by the ImageNet-C dataset creators.
- The relationship between contrastive learning and classification in Section 4.1 is, I think, well-known, e.g. Wu et al. (2018) motivate their work in this way. The statement that “the SimCLR criteria is equivalent to a negative crossentropy [sic] loss over the output of a classifier which predict the index of a given image with respect to a random transformation” does not seem quite correct because this classifier would conventionally have different weights for each example rather than using another example’s representation as the weights. I also do not understand the meaning of the brackets in Eq. 2.
- There are a couple of papers that explore the use of a projection head in supervised learning that are not cited but probably should be. Off the top of my head, I know of Wang et al. (2022) and Sarıyıldız et al. (2023).
- “Guillotine Regularization” is a fun name, but at this point a new name for the use of a projection head is unlikely to get widely adopted, and I am not entirely convinced that it is a form of regularization.

References:

Chen, T., Kornblith, S., Norouzi, M., & Hinton, G. (2020, November). A simple framework for contrastive learning of visual representations. In International conference on machine learning (pp. 1597-1607). PMLR.

Goldfeld, Z., Berg, E. V. D., Greenewald, K., Melnyk, I., Nguyen, N., Kingsbury, B., & Polyanskiy, Y. (2019). Estimating Information Flow in Deep Neural Networks. In International Conference on Machine Learning.

Sarıyıldız, M. B., Kalantidis, Y., Alahari, K., & Larlus, D. No Reason for No Supervision: Improved Generalization in Supervised Models. In International Conference on Learning Representations.

Wang, Y., Tang, S., Zhu, F., Bai, L., Zhao, R., Qi, D., & Ouyang, W. (2022). Revisiting the transferability of supervised pretraining: an mlp perspective. In Proceedings of the IEEE/CVF Conference on Computer Vision and Pattern Recognition (pp. 9183-9193).

Wu, Z., Xiong, Y., Yu, S. X., & Lin, D. (2018). Unsupervised feature learning via non-parametric instance discrimination. In Proceedings of the IEEE conference on computer vision and pattern recognition (pp. 3733-3742).

---

> ### Author Response · Authors · 2023-03-02
> **Answer to Reviewer hcdo: Strengths And Weaknesses**
>
> Thank you for your review. We invite you to read the general answer to reviewers as well as the updated pdf.
>
> > The intro lists as one of the paper's contributions "To formalize and motivate theoretically the common trick of using a projector in Self-Supervised Learning" but I do not believe it makes a meaningful contribution here. ...
> > Since Chen et al. (2020)'s justification for the projection head is informal, the Information Theoretical Motivation could still be a contribution of interest to TMLR's audience if it were correct ...
>
> You're right, we removed the Theoretic Motivation part since it's not a contribution of this paper (it was also misleading and incorrect). We updated the list of contributions in the introduction and provide a new high level intuition that don't assume anything about the flow of information going through the network.
>
> > The paper seems to provide two contradictory descriptions of what is shown in Figure 3c....
> I wonder whether how robustness measured on ImageNet-C at the backbone of a network trained with a projection head compares with robustness of a standard ResNet-50 with no projection head ...
>
> We corrected it, the probes are trained on ImageNet, the evaluations of the probe is performed on ImageNet-C. We added the mCE score in Table 1. We used the following spreadsheet https://docs.google.com/spreadsheets/d/1RwqofJPHhtdRPG-dDO7wPp-aGn-AmwmU5-rpvTzrMHw from https://github.com/hendrycks/robustness to compute the mCE. The score reported in the spreadsheet (unormalized mCE) for a Resnet50 without projection head is 60.8 while our Resnet50 at the last projection layer has 59.0 while giving 63.3 at the trunk layer.
>
> > The relationship between contrastive learning and classification in Section 4.1 is, I think, well-known, e.g. Wu et al. (2018) motivate their work in this way... I also do not understand the meaning of the brackets in Eq. 2.
>
> We wanted to use this as a motivation for section 4 but you're right, this result is not new. In consequence, we removed Section 4.1 and reference directly Wu et al. (2018) at the beginning of the section.
>
> > There are a couple of papers that explore the use of a projection head in supervised learning that are not cited but probably should be. Off the top of my head, I know of Wang et al. (2022) and Sarıyıldız et al. (2023).
>
> Thank you for these very relevant references. We added those in the updated version of the paper.
>
> > “Guillotine Regularization” is a fun name, but at this point a new name for the use of a projection head is unlikely to get widely adopted, and I am not entirely convinced that it is a form of regularization.
>
> The point of using another name was mostly because the action of throwing away the projection head is misleading. If we look at Figure 3, the best layer for a given factor of variation might be before the backbone. When looking at Figure 5, the best layer might be the first projector layer. When looking at Figure 11 with the VIT experiment, we see that the best layer is also inside the projector not at the backbone ! The projection head name trick implies that the backbone layer is always the best and that the entire projector should be discarded. While using a more general name like Guillotine really emphasizes that what matters is chopping layers without any a priori about what constitutes the head of the network (It can be one part of the projector, the entire projector or the projector + other layers). So we wanted to have a more meaningful word that separates the technique (of chopping layers) from the architecture of the network (which can be arbitrary, there might be some instances or some architecture for which we might even not need to add a projector if we cut deep enough in the network). Concerning regularization, we use it in a broad context which is a way to avoid overfitting on the pretext task. When looking at Figure 4) b), we clearly see that this technique can be used to avoid overfitting. However we agree that it’s not a regularization when looking at a traditional ML definition since usually the regularization will reduce the variance at the cost of increasing the bias, in our case, we show that it might reduce the bias with respect to the optimal downstream function but we don’t assume anything about the variance. If the word regularization is still a concern, we can remove it from the paper (including the title).

---

> > ### Author Response · Authors · 2023-03-02
> > **Answer to Reviewer hcdo: Requested Changes**
> >
> > > Unless there is a way to fix the theoretical claims, they should be stated only as intuition or removed, and they should not be claimed as a contribution.
> >
> > We removed them and stated the remaining motivations as intuition.
> >
> > > The description of the experiments in Figure 3c should be made consistent. Ideally the paper would also report mCE for each layer in addition to performance on the individual corruptions.
> >
> > We present the mCE for each layer in Table 1. We removed the individual corruption figure since it doesn’t give any additional information than the mCE. However we could add the performance on individual corruptions in the appendix if needed.
> >
> > > If the authors want to keep Section 4.1, it should cite previous work that has made similar observations, but in my opinion it does not contribute much to the paper and could be removed.
> >
> > We removed 4.1.
> >
> > > The papers above should be cited.
> >
> > Done !
> >
> > > I think it's fine to use "Guillotine Regularization" as a catchy title but I'd suggest minimizing the use of the term elsewhere. It's unnecessary and potentially confusing to rename the use of a projection head given that it is already a common technique. The abstract should also explicitly refer to an "MLP projection head" or "MLP projector" since that is what people commonly call this trick in previous literature.
> >
> > We update the abstract and use the name of MLP projection head when it’s a better fit. However, one of the contributions of this work is to highlight that the fact of throwing away the projector head is misleading since it implies that the entire projector should be removed after training. In this paper we show that the optimal layer to use highly depends on the training setup as well as which downstream task to evaluate on. So we think it's important for the community to distinguish between the architectural modification of adding a projector and the action of cutting layers (which is independent of the presence of a projector or not). We hope that the name Guillotine might facilitate making such a distinction.

---

> > > ### Comment · Reviewer_hcdo · 2023-03-26
> > > **Response**
> > >
> > > Dear Authors,
> > >
> > > Thank you for making these changes, and sorry for taking so long to respond to your comments.
> > >
> > > I think it is well-known that it is sometimes better not to throw away the entire MLP projection head. [SimCLR v2](https://proceedings.neurips.cc/paper/2020/hash/fcbc95ccdd551da181207c0c1400c655-Abstract.html) previously made the observation that the optimal layer to transfer is sometimes in the middle and this should be mentioned someplace. In general, I still find the way the paper is framed to be a little weird. In my opinion, the projection head is not "usually skimmed-over" (as in the abstract); there is no "preconceived idea that we should through [sic] away the entire projector in SSL" (as in the abstract); and throwing the projection head away is not "misleading" (as on p.2) because, for many downstream tasks including ImageNet linear classification, it is actually the optimal thing to do. I think there are enough new experiments here that this paper will be interesting to TMLR's audience, but I wish the authors would tone down the novelty claims; I personally find them distracting and unnecessary.
> > >
> > > The revised submission also places a lot of emphasis on "bias" without describing exactly what is meant. Usually bias (in the ML/stats sense) relates two functions to each other but the paper talks about functions having "a low bias with respect to the optimal
> > > readout downstream task" or "the bias of the readout function for the downstream task," which is harder for me to understand.
> > >
> > > There are also some minor typos in the added text; I see:
> > > - "through" -> "throw" (in abstract)
> > > - "Mostly because the optimal layer..." (p. 2, this is a sentence fragment that should be combined with the previous sentence)
> > > - "alignement" (p. 2)
> > >
> > > I will recommend acceptance if the authors commit to fixing these issues in the final version.

---

> > > > ### Author Response · Authors · 2023-04-03
> > > > **Answer to Reviewer hcdo**
> > > >
> > > > Dear reviewer,
> > > >
> > > > Thank you for the follow up and the helpful suggestions. We updated the paper with the requested changes which are shown in red in the pdf. Please let us know if there is anything else.
> > > >
> > > > Concerning our discussion of bias, we were indeed speaking about the bias between two functions, and apologize if this wasn’t clear. We argued that the function that produces the final representation i.e the full network trained on the pretext task, will logically have the smallest possible bias with respect to the optimal function that solves the pretext task. Then we compared the function that is obtained by tuning the readout on the downstream task, to the optimal downstream task function. Since cutting layers creates “different” sub-networks, tuning a readout on these “sub-networks” yields predictive functions that could have different biases with respect to the optimal downstream task function.
> > > >
> > > > However, after trying to clarify the paragraph about this bias perspective, we came to the conclusion that it might just be better to remove this discussion entirely. One of the main points in the reviews was that our theoretical justification was too handwavy and confusing, which was true. So we first tried to replace it with a better high level explanation of what may be going on, based on the notion of bias, in the hope that it could provide some clearer intuition. But the argument does remain handwavy, since we cannot directly quantify this bias. So we think it is probably better to just remove our attempt at providing theoretical justification or motivation for the technique, since the core of our contribution is, as you rightly highlighted, essentially empirical.
> > > >
> > > > We hope that these changes will address your concerns.
> > > >
> > > > Thank you again for your time.

---

### Author Response · Authors · 2023-03-02
**General answer to all reviewers**

We would like to thank all reviewers for the valuable comments and suggestions. They have significantly helped us to improve the quality of the paper. One of the common criticisms was on the theoretical motivation which was limited. We shouldn’t have presented it as a contribution since its goal was only to provide a high level intuition about why the method should work. Furthermore, as noted by the reviewers, the theoretical explanation we presented was limited and misleading since it’s difficult to quantify the flow of information in deep networks. Because it isn’t the focus of the paper, we decided to remove the considerations around mutual information. Instead we replaced it by a high level illustration which doesn’t assume anything about the amount of information at a given layer but that assumes that the readout functions trained on different representations (at different layers) will have a different bias with respect to the optimal readout downstream function. In consequence, the best we can do is to train readout functions at different layers and measure the performance in order to determine which one has the lowest bias. We hope that this new motivation will satisfy the reviewers.

A second important point was around the use of the name Guillotine Regularization instead of the projector head trick in SSL. These comments helped us to realize that the paper lacked clarity concerning our motivation for introducing a new name. In the revised paper, we argue that using “projector head trick” is misleading in SSL since it assumes that you should throw away the entire projector after training. The main motivation to use the name Guillotine was to distinguish between the technique of cutting layers (which is used since a long time outside SSL) from the architectural modification which is the addition of a projector. In the new version of the paper, we highlight how much the optimal layers to cut are different depending on the training setup, data and downstream tasks. In consequence, we strongly encourage SSL researchers and practitioners to stop throwing away the entire projector after training and instead determine through experiments which individual layers to throw away for each downstream task. In fact, in an ideal scenario, the layer that should be used for each downstream evaluation should be part of an hyper-parameter search.

We also removed section 4.1 since we didn’t consider it as a contribution of this paper. Lastly, we move “the robustness to hyper-parameters” to section 3. The main message we wanted to convey was to show that the performances at the projector and at the representation level aren’t always correlated. We can have increasing performances at the backbone layer while having decreasing projector performances. We hope that this new explanation is more clear than the previous one.

We invite the reviewers to read our updated pdf (the modifications are in blue), here is the changelog:

- Remove the mutual information motivation and replace it by Figure 2 that gives a high level intuition about how different readouts have different biases.

- Show that “throwing away the projector in SSL” is misleading since the optimal layer to use depends on multiple factors.

- Add a section showing that different training setup might lead to different performances across readout functions at different layers (Figure 4 a) and b)).

- Contrast the downstream supervised part with the downstream ssl part in Figure 5 which show clearly that the optimal layer to use is downstream task dependent.

- Move Figure 6 in Section 3.

- Replace Imagenet-C figure by a table with the mCE scores.

- Remove subsection 4.1

- Add visualization of the information that is retained across layers for section 4.

Please let us know if there is anything else to clarify or add as an experiment.
Thank you again for your time.

---

> ### Author Response · Authors · 2023-03-15
> **Changelog 2**
>
> Following recommendations from Reviewer JDve, we decided to:
>
> - Remove Figure 2
> - Update the paragraph around Table 1.
> - Update the title to highlight our focus on Self-Supervised Learning.

---

> > ### Author Response · Authors · 2023-04-03
> > **Changelog 3**
> >
> > Following recommendations from Reviewer hcdo, we decided to:
> >
> > - Tone down the novelty claims in the abstract and the introduction
> > - Add SimCLR v2 discussion in the related work.
> > - Remove discussion on the bias in section 3

---

### Decision · Action_Editors · 2023-04-30

**Recommendation:** Accept with minor revision

**Comment:**

The authors provide a series of empirical studies of the properties of the projection head of self-supervised learning (SSL) methods, which AE think is useful for the TMLR SSL community. There are several concerns of reviewers initially (including over-claims, limited theoretical parts, lack of clarification/motivation), but the authors succeed in revising the draft for resolving them. Hence, all reviewers suggested acceptance finally. AE thinks that the results in this paper are not super novel, but some of them can be useful for SSL practitioners. Nevertheless, for preparing the camera-ready draft, AE strongly recommends the authors going through the draft again from the beginning to the end, polishing it further, e.g., fixing typos and toning-down some strong sentences for avoiding some potential confusion.

**Audience:**

Yes

**Claims And Evidence:**

Yes